

# TERRESTRIAL AND MARINE PLASTIC POLLUTION OUTLOOK IN THE MEDITERRANEAN REGION: A BOX-MODEL APPROACH BASED ON OECD POLICY SCENARIOS.

Théo SEGUR[1] and Jeroen E. SONKE[1]

[1]Géosciences Environnement Toulouse, CNRS/IRD/Université Paul Sabatier Toulouse 3, Toulouse, France
**Correspondence:** Théo SEGUR (theo.segur@get.omp.eu) and Jeroen E. SONKE (jeroen.sonke@get.omp.eu)

**Abstract.** Plastic pollution in the Mediterranean region and Sea raises serious concerns for ecosystem and human health. Plastic dispersal from Mediterranean watersheds in Southern Europe, Northern Africa and Middle-East, and Nile basin is complex due to the different (mis-)managed waste streams, population dynamics and climate. In this study, an environmental plastics mass budget and box-model is proposed for the Mediterranean region based on recent observations. We use this model to explore
plastics dispersal under different OECD plastic production and waste management policy scenarios toward the end of the 21st century. We find that the current Mediterranean marine plastic stock (sea surface, water column, sandy beach and sediments) of 7 million metric tons (Mt, median, IQR 3-15 Mt) in 2015 constrains continental plastic runoff to 0.31 Mt y$^{-1}$ (median, IQR 0.14-0.57 Mt y$^{-1}$). The total marine plastics stock would increase 4-fold by 2060 under a business-as-usual scenario, reaching 26 Mt (median, IQR 13-48 Mt). Implementation of the OECD Global Ambition policy scenario, that targets near-zero new
plastics waste leakage, would not significantly lower this stock (25 Mt, median, IQR 12-44 Mt) by 2060. This is because marine litter remote sensing observations attribute most, 0.27 Mt y$^{-1}$ (88%), of recent plastic runoff to Southern Europe, where high rainfall will continue to mobilize legacy plastic waste from land to sea, regardless of low leakage targets. About 1.5% of all Mediterranean legacy plastic waste reached the marine environment, meaning that most plastic waste still resides on land (361 Mt, 76%). Moreover, in the marine environment, 83% of plastic mass resides in shelf sediments (median 6 Mt, IQR 2-14
Mt), which are fragile ecosystems that host most of the Mediterranean Sea biodiversity, and are not easy to clean up. This underlines the necessity to address upstream legacy plastic waste on land. Land-based remediation scenarios modelled here show that total plastic runoff from land to sea can be reduced 2-fold (0.22 Mt y$^{-1}$, IQR 0.16-0.27 Mt y$^{-1}$) compared to the business-as-usual scenario in 2060, and significantly reduce the total plastic stock in the marine environment.

## 1   Introduction

Plastic pollution is emerging as a global concern for the environment and human health (IUCN, 2023). Global primary plastic production has reached more than 400 Mt (1012 g or millions of metric tons) per year in 2016 (Geyer et al., 2017), and continues to grow by more than 4% every year since. For the year 2015, it was estimated that 8 300 Mt of primary plastic had





been produced since 1950, of which 2 500 Mt are still in use and 5 800 Mt were discarded (Geyer et al., 2017). Around 9% of this discarded waste has been recycled, 12% incinerated, and the remaining 79% has been disposed of in landfills, dumpsites, or littered in nature (Geyer et al., 2017). Due to their slow biodegradability, disposed plastics have a long persistence in the environment (Chamas et al., 2020), creating important "legacy" plastic waste stocks estimated around 4 900 Mt globally in 2015 (Geyer et al., 2017; Sonke et al., 2022). Plastic items are also very mobile due to their relatively low density and buoyancy, and can travel long distances by rivers and ocean currents. It is estimated that rivers carry from 1 to 6 Mt of plastic each year globally (González-Fernández et al., 2023). Some studies propose even higher plastic flux, up to 14 Mt y$^{-1}$ (Jambeck et al., 2015; Sonke et al., 2024). Another vector of large-scale dispersion of small microplastics (<0.3mm) is the atmosphere, where plastic particles are emitted from land (as agricultural dust, tire and brake wear particles, urban-industrial dust) (Brahney et al., 2021) and from the sea (Allen et al., 2022, 2020), and undergo long-range transport (Allen et al., 2021b; Tatsii et al., 2024; Xiao et al., 2023) before deposition to remote land surfaces. As a consequence, plastics are now an ubiquitous material in the environment, whose presence has been reported in freshwater systems and oceans (Schwarz et al., 2019), marine sediments (Barrett et al., 2020), atmosphere (Wang et al., 2023), as well as polar regions (Bergmann et al., 2022), mountains and remote soils (Allen et al., 2021a). Plastics exert various pressures on the environment, varying, among others factors, with their size. Macroplastics (items greater than 5mm) have been shown to entangle various marine animals, or block their digestive tract (Browne et al., 2015; Gregory, 2009). Macroplastics floating at the ocean surface can also serve as refuge for invasive or non-indigenous species and contaminate new regions through drifting and beaching (Aliani and Molcard, 2003; Miralles et al., 2018; Rech et al., 2018). Microplastics (< 5 mm) can have a wide range of potentially adverse effects on biota, from genetic expression to physiological or behavioural changes (Tuuri and Leterme, 2023). Microplastics can release toxic additives (e.g. endocrine disruptors, Bisphenol A), or serve as vectors for toxic chemicals such as hydrophobic organic compounds (polychlorinated biphenyls, polyaromatic hydrocarbons...), organochlorine pesticides, or trace metals (Verla et al., 2019). The Mediterranean Sea is considered as a hotspot for biodiversity, with more than 17 000 species identified, among which 20% are endemic (Coll et al., 2010). This biodiversity is linked to an important diversity of habitats and ecosystems. These ecosystems are threatened by the intense human activity occurring near its coast. Around 190 million people were living less than 50 km from the Mediterranean coastline in 2020 (UN, 2022), to which must be added the 360 million tourists per year (in 2017), accounting for 27% of international tourism (UNEP/MAP, 2020). Furthermore, the Mediterranean Sea is crossed by 30% of the international shipping traffic (Bolo and Préville, 2020), and presents highly urbanized and industrialized zones near its coast (Poulos, 2020; UNEP/MAP, 2020). As a consequence of this intense human activity, the Mediterranean Sea is considered as one of the most affected environments by marine litter (Fossi et al., 2020; UNEP, 2021). Numerous studies have tried to estimate the quantity of plastic floating at the Mediterranean Sea surface. Pedrotti et al. (2022) have estimated 660 metric tons of floating plastics, while Cózar et al. (2015) estimated between 1000 and 3000 metric tons. Simon-Sánchez et al. (2022) reviewed and reported concentrations in sediments (300 items kg$^{-1}$) and beaches (60 item kg$^{-1}$), insisting on the high uncertainties and variability between studies. Demographic trends (UN, 2022), plastic usage (OECD, 2022) and MMPW generation (Lebreton and Andrady, 2019; OECD, 2022) are widely different between Mediterranean Sea sub-regions such as Southern Europe, Northern Africa and Middle-East and the Nile basin region. Several studies have used these indicators to estimate the global



plastic runoff from land to sea (Lebreton et al., 2017; Meijer et al., 2021; Nyberg et al., 2023), and show similar geographical distribution of plastic runoff to sea in the Mediterranean region (dominated by Northern Africa and Middle-East). However, a recent satellite remote sensing study of Mediterranean Sea litter by Cózar et al. (2024) highlighted the close relationship between marine litter occurrence and heavy rainfall events, pointing at Southern Europe as the largest macroplastic source to the Mediterranean Sea. Several modelling studies have focused on the plastic issue in the Mediterranean region (Boucher and Billard, 2020; Kaandorp et al., 2020; Liubartseva et al., 2018; Pedrotti et al., 2022; Tsiaras et al., 2021). Kaandorp et al. (2020) used a 2D Lagrangian model to fit plastic runoff to sea, based on observed floating plastic concentrations in the Mediterranean Sea. In their box-model, Boucher and Billard (2020) created one of the first multi-compartmental studies assessing plastic distribution in the Mediterranean Sea and explored scenarios toward plastic pollution reduction by 2040. However, no plastic budget or modelling studies have integrated the Mediterranean marine, terrestrial and atmospheric compartments together, and explored future plastic policy scenarios toward the end of the century based on international agencies' recommendations. This study investigates the fate of plastic waste in the Mediterranean Sea catchment across various environmental compartments (terrestrial, sea surface and water column, shelf and deep sediments, beaches and atmosphere). We first propose a Mediterranean plastic mass budget for the year 2015, allowing us to calibrate a mass balance box-model and propose a top-down estimation of plastic runoff from land to sea. The Mediterranean Sea catchment is subdivided into 3 regions (Southern Europe, Northern Africa & Middle-East and the Nile River basin), each having its own plastic waste forcing curve derived from region-specific demography, plastic waste generation and mismanaged fraction. The box-model is then used to estimate the evolution of historical plastic stocks in the Mediterranean basin between 1950 and 2015, and explore several future plastic policy scenarios based on the Organization for Economic Co-operation and Development (OECD) outlook toward 2060 (OECD, 2022).

## 2 Methods

### 2.1 Study area

This study focuses on plastic contamination in different marine, terrestrial and atmospheric compartments in the Mediterranean region. The Mediterranean Sea catchment is subdivided into 3 regions shown in Fig.1: Southern Europe (S. Europe), regrouping EU and non-EU countries; North Africa and the Middle East (N. Africa & M. East) and the Nile River catchment (Nile basin). The boundaries between regions follow river catchment delimitations, provided by the HydroSHEDS database (Lehner and Grill, 2013).

### 2.2 Model structure & parametrization

In order to assess the plastic contamination in different environmental compartments, a first-order 1D mass transfer model is developed based on the previously published work by Sonke et al. (2022, 2024). This box-model, presented in Fig.2, is the representation of the plastic stock in 9 environmental compartments (sanitary landfills, mismanaged terrestrial pool (MMPW), remote soils, sea surface, water column, shelf and deep sediments, beaches and atmosphere) expressed in millions of metric



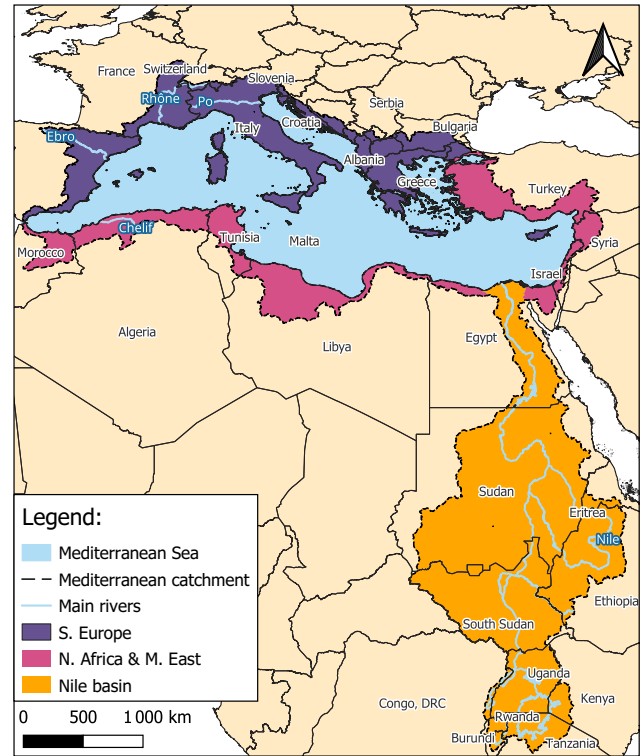

**Figure 1.** (single column) Map of the study area and the 3 geographical plastic source regions considered. Mediterranean Sea catchment outline is provided by the HydroSHEDS database (Lehner and Grill, 2013). Region boundaries follow river catchment delimitation.

tons (Mt). Compartments are subdivided into boxes according to plastic size category they are expected to contain macroplas-

tics (P) which characteristic length L is greater than 5 mm, large microplastics (LMP) defined as 0.3 mm $\leq$ L<5 mm and small microplastics (SMP) defined as 1 $\mu$m $\leq$ L<0.3 mm. We will use the general term plastic (Ptot) to refer to the total amount of plastic (P + LMP + SMP), and microplastic (MP) for the fragments less than 5 mm in size (LMP + SMP). In total, 35 boxes are considered in this model. Boxes are linked together by mass transport of plastic of two types : (1) fluxes that represent movement of mass between compartments through runoff, gravity or wind (thin arrows in Fig.2) and (2) plastic fragmentation that

transfers mass from a larger plastic size category to a smaller size category (P to LMP to SMP) within the same compartment (bold arrows in Fig.2). Fragmentation is caused by (photo-)oxidation, structural fragilization and mechanical abrasion at a rate of approximately 3% per year (Chamas et al., 2020; Lebreton et al., 2019; Sonke et al., 2022, 2024). This rate is applied in the MMPW terrestrial pool, the sea surface and the sandy beaches compartments.

      Plastic waste enters the model via the 3 terrestrial regions. Each waste flux is immediately divided between recycled, inciner-

ated, discarded to sanitary landfills and mismanaged. In this study, sanitary landfills are considered as "plastic-tight", meaning no leakage of plastic to the environment from this compartment is assumed over the time scale studied (1950-2100). The MMPW compartment corresponds to the terrestrial environment directly polluted by human activities, including non-sanitary

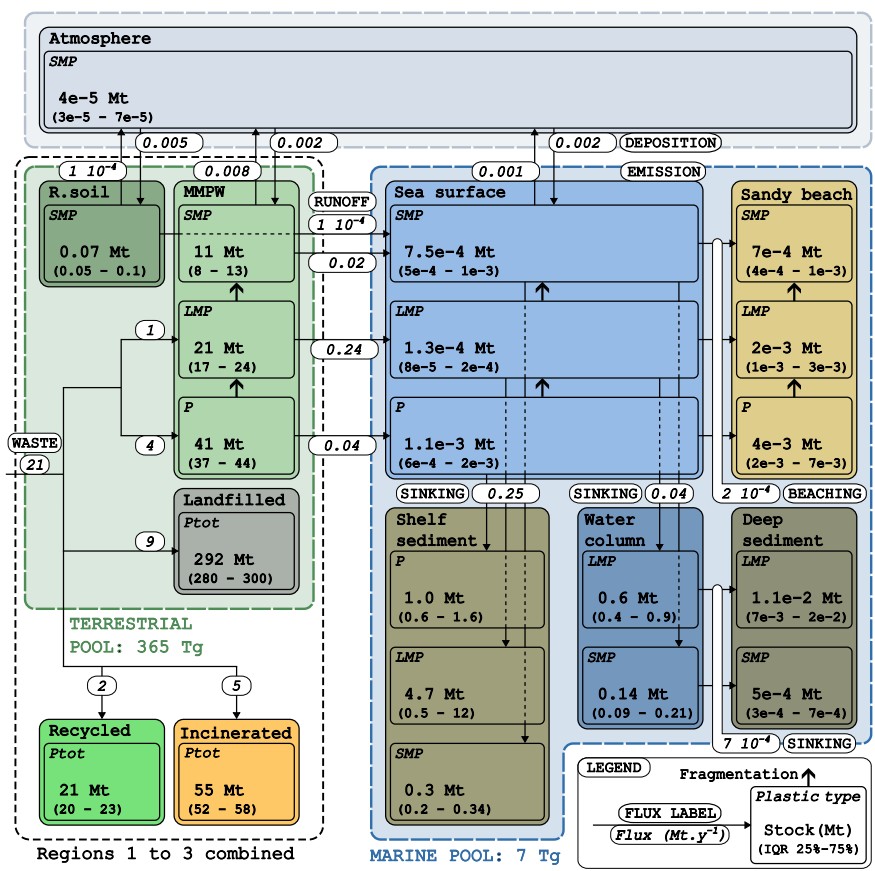

**Figure 2.** (double column) Structure of the box-model and model estimation of plastic stocks and fluxes for the reference year 2015. Each box is associated with a plastic size category (P ≥ 5mm, 0.3mm ≤ LMP < 5mm and 1$\mu$m ≤ SMP < 0.3mm) and its corresponding modelled median stock in millions of metric tonnes (Mt). The uncertainties provided are interquartile range (Q25-Q75). Numbers in white labels are the annual plastic flux between boxes (in Mt y$^{-1}$). When the label stretches across multiple arrows it indicates the combined flux of these arrows. Bold arrows represent degradation of plastic items to smaller size categories (P to LMP to SMP) under UV and mechanical abrasion. Note that the 3 modelled terrestrial regions are combined on the left panel. "R. soil" stands for "remote soil surfaces" and "MMPW" for "terrestrial mismanaged plastic waste".





dumps, urban-industrial areas, agricultural soils, freshwater lakes and wetlands. From there, the smaller plastic fraction (SMP here) can be transported through wind erosion to the atmosphere (Brahney et al., 2021) or through continental runoff of P,

LMP, SMP to the Mediterranean sea surface pool (van Emmerik and Schwarz, 2020; Jambeck et al., 2015; Lebreton et al., 2017; Meijer et al., 2021). Atmospheric SMP can be deposited on the sea surface or on land (Allen et al., 2022; Brahney et al., 2021), both to the terrestrial MMPW pool (urban area and agricultural land) or to remote soils (mountains, deserts, remote land surfaces...). From the sea surface, plastics items can be deposited on beaches (Hinata et al., 2017), and SMP can be emitted to the atmosphere through sea spray (Allen et al., 2020; Brahney et al., 2021; Shaw et al., 2023; Wu et al., 2021). MP can

also sink to the shelf sediments and to the deep-water column due to biofouling and/or gravity. Additionally, we consider that P is able to sink to the shelf sediments near the shore. Finally, MP can further sink from the deep-water column to the deep sediments (see Fig.2).

Plastic mass transport between boxes is approximated to be first order, where the plastic flux $F_{A \to B}$ [Mt y$^{-1}$] from box $A$ to box $B$ is proportional to the plastic mass $M_A$ [Mt] in box $A$:

$$F_{A \to B} = k_{A \to B} \times M_A \tag{1}$$

Where $k_{A \to B}$ [y$^{-1}$] is the proportionality coefficient defining the plastic mass flux from box $A$ to box $B$. The set of all k-values are the parameters of the model and are kept constant for all the integration period. The variation of plastic stock in box $A$ at time $t$ is the sum of all incoming fluxes minus the sum of all exiting fluxes from box $A$ at time $t$:

$$\frac{dM_A}{dt} = \sum_{i \in I} F_{i \to A} - \sum_{j \in J} F_{A \to j} = \sum_{i \in I} k_{i \to A} \times M_i - \sum_{j \in J} k_{A \to j} \times M_A \tag{2}$$

Where $I$ is the set of boxes upstream of box $A$, and $J$ is the set of boxes downstream of box $A$. The solution to this system of ordinary differential equations gives the plastic stock of each box at any time in the integration interval. Solutions were computed using an implicit Runge-Kutta method (Radau IIA family of order 5) compiled in a Python script ($Python 3.11.5$) available in SI. To evaluate uncertainties, a Monte-Carlo approach is implemented where model coefficients k are randomly drawn from their assigned probability (uncertainty) distribution across 1000 model runs. Probability distribution of all k-values

is available in Table S1 & S2.

### 2.3 Model forcing

Plastic is entering the model circulation through waste generation in the 3 terrestrial regions. To estimate the quantity of plastic waste generated at time $t$ for each region $R$, forcing curves $PW(R,t)$ [Mt y$^{-1}$] were computed using Eq. (3).

$$PW(R,t) = PCPW(R,t) . \sum_{c \in R} f_{catch}(c) \times Pop(c,t) \tag{3}$$

$PCPW(R,t)$ [Mt capita$^{-1}$ y$^{-1}$] is the per capita plastic waste generation rate derived from OECD (Fig.3.b & Fig. S1) for the three regions (OECD, 2022). $Pop(c,t)$ [capita] is the total population of the country $c$, obtained from UN demographic data (UN, 2022). Historical estimates are used from 1950 to 2022, and medium fertility variant estimates from 2023 to 2060.



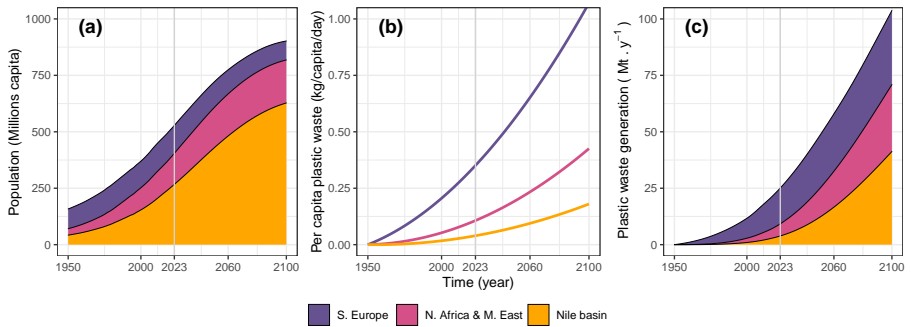

**Figure 3.** (double column) Population, per capita plastic waste and total plastic waste generation projections by region. Colours correspond to the terrestrial regions. (a) Stacked regional population derived from UN statistics (historical estimates from 1950 to 2022, and medium fertility variant estimates from 2023 to 2060) and georeferenced population data (WorldPop, 2020). (b) Per capita plastic waste generation rates under Business-as-Usual (BAU) scenario, derived from OECD estimations (OECD, 2022) (see Fig. S1 for details). (c) Stacked annual plastic waste generation per region under BAU scenario, corresponding to the combination of graph A and B according to Eq. (3).

$f_{catch}(c)$ [-] is the fraction of the population in the country c that lived in the Mediterranean catchment in 2015. fcatch is computed using georeferenced population data provided by the WorldPop database (WorldPop, 2020), adjusted to match the
corresponding UN data (UN, 2022). Using GIS software QGIS (version 3.28.4), the population living inside the Mediterranean catchment is computed for each country and divided by the total population of the country for the year 2015 to obtain $f_{catch}(c)$. Further testing showed that this fraction can be considered constant from 2000 to 2020, and therefore is assumed constant for the whole time frame studied as a first approximation (see Fig. S2 for details). A 5% uncertainty is introduced to these forcings following Geyer et al. (2017) estimations.

**2.4   Mediterranean plastic observations**

In order to calibrate our model, plastic stocks and fluxes in the Mediterranean study region were assessed based on an extended literature review of recently (2015-2024) published articles. The year 2015 is chosen as reference for calibration. When reported in terms of the number of plastic items per size category, the numeric concentrations were converted into mass concentrations assuming an average ellipsoid particle volume $V = L^3/10$ (L is the reported average length of the size category) and an average
plastic item density of $\delta_P = 1 \ g \ cm^{-3}$ (Kooi and Koelmans, 2019).

**2.4.1   Sea surface**

The Mediterranean Sea surface is the most documented compartment studied. 2 studies directly provided an estimation of the total surface plastic stock (Boucher and Billard, 2020; Pedrotti et al., 2022). Other studies (Baini et al., 2018; Cincinelli et al., 2019; Cózar et al., 2015; Simon-Sánchez et al., 2022) provided average plastic mass concentrations [g km$^{-2}$] that were
multiplied by the Mediterranean Sea surface area ($2.5 \ 10^6 km^2$) to obtain an estimation of the plastic stock in the sea surface.



Among these studies, one review (Cincinelli et al., 2019) provided numeric concentration [item km$^{-2}$] that were converted to mass concentration [g km$^{-2}$] using the item to mass conversion formula elaborated by Cózar et al. (2015) Cózar et al. (2015) for floating plastics:

$$log(M\ [g\ kg^{-2}]) = 1.22 \times log(N\ [item\ kg^{-2}]) - 4.04 \qquad (4)$$

The estimations of Kaandorp et al. (2023) of global floating P and LMP, and the estimation of Sonke et al. (2024) of global LMP were downscaled to the Mediterranean sea surface in order to account for the new higher macroplastic quantification. Summary of the plastic stock at the Mediterranean Sea surface are presented in Table 1, and amount to 8.9 10$^{-4}$ Mt (median, IQR 6.6 10$^{-4}$-4.0 10$^{-3}$ Mt) of floating P and 2.7 10$^{-4}$ Mt (median, IQR 1.8 10$^{-4}$-3.6 10$^{-4}$ Mt) of floating LMP. Note that no reliable SMP data was found for the Mediterranean region (see section 2.4.5).

**Table 1.** Literature review of plastic stock in the Mediterranean Sea surface P ≥ 5mm, 0.3mm ≤ LMP < 5mm and 1$\mu$m ≤ SMP < 0.3mm. If available, the uncertainties are given with the following methodology: [1] 95% confidence interval, [2] interquartile range, [3] mean ± 1$\sigma$

| Reference | Sea surface stock [tonnes] | |
| --- | --- | --- |
| | P | LMP |
| Kaandorp et al. (2023) | 12 700 | 650 |
| Cózar et al. (2015) | 1 067 (756-2969) | |
| Boucher and Billard (2020) | 705 (288-1840)[2] | |
| Pedrotti et al. (2022) | 504 (293-984)[1] | 156 (125-202)[1]135 (47-223)[1] |
| Baini et al. (2018) | | 104 ± 172[3] |
| Cincinelli et al. (2019) | | 383 (227-858)[2] |
| Sonke et al. (2024) | | 277 |
| Simon-Sánchez et al. (2022) | | 252 (70-914) |
| Median (IQR 25-75%) | 886 (655-3975) | 265 (180-357) |

### 2.4.2 Sandy beaches

In their study, Boucher and Billard (2020) reviewed beach litter collection data and directly estimated the stock of plastic in Mediterranean sandy beaches (0.001 Mt, Table 2.). In their review, Haarr et al. (2022) reported beached P densities of 5650 kg km$^{-2}$ (median, IQR 1500-17700 kg km$^{-2}$) for the Mediterranean coastline. We multiplied the reported P density by the estimated sandy beach area in the Mediterranean basin, assuming a coast length of 47 200 km among which 46% is sandy beaches (Poulos, 2020) and an average beach width of 50 m, to obtain an estimate of 0.006 Mt (Table 2).




**Table 2.** (single column) Literature review of plastic stock in the Mediterranean sandy beaches, water column and shelf sediments. The reported concentration, if provided, was transposed to Mediterranean Sea stock (Med. Stock) using item to mass conversion ($V = L^3/10$) and/or surface extrapolation.

| Box | Reference | Reported concentration | Med. stock [Mt] |
|---|---|---|---|
| Sandy beach P | Haarr et al. (2022) | $5.6\ 10^{-3}$ | 0.006 |
| | Boucher and Billard (2020) | | 0.001 |
| Water column LMP | Baini et al. (2018) | $2.6\ 10^{-1}$ LMP m$^{-3}$ | 1.2 |
| | Lefebvre et al. (2019) | $2.3\ 10^{-1}$ LMP m$^{-3}$ | 0.02 |
| Shelf sediment P | Sonke et al. (2024) | $1.1\ 10^{2}$ Mt | 1.7 |
| | Cau et al. (2024) | | 1.5 |
| | Zhu et al. (2024) | | 0.5 |
| | Zhu et al. (2024) | | 0.2 |
| Shelf sediment LMP | Alomar et al. (2016) | $1.9\ 10^{3}$ g m$^{-3}$ WW | 86.1 |
| | Fastelli et al. (2016) | $3.3\ 10^{2}$ g m$^{-3}$ WW | 16.0 |
| | Kazour et al. (2019) | $9.4\ 10^{-3}$ g m$^{-3}$ WW | 4.2 |
| | Mistri et al. (2018) | $7.3\ 10^{-1}$ g m$^{-3}$ WW | 0.04 |
| | Filgueiras et al. (2019) | $9.7\ 10^{-2}$ g m$^{-3}$ WW | 0.001 |

### 2.4.3 Water column

Two sampling studies were reviewed to estimate the water column plastic stock (Baini et al., 2018; Lefebvre et al., 2019). Both of them only sampled down to 100 m depth, which is not representative of the deep Mediterranean water column, whose average depth is 1547 m (Poulos, 2020). The concentrations found by these studies were therefore corrected using model results by (Tsiaras et al., 2021) that investigated the vertical mixing of MP in the Mediterranean Sea. According to their result, 70% of the MP mass sunk under 100 m depth, which allowed us to convert the 0-100 m plastic stock to 100-1547 m (1.2 Mt and 0.02 Mt for Baini et al. (2018) and Lefebvre et al. (2019) respectively, see Table 2)

### 2.4.4 Shelf and deep sediments

We reviewed 8 studies on plastics in shelf sediments (depth < 100m) (Alomar et al., 2016; Cau et al., 2024; Fastelli et al., 2016; Filgueiras et al., 2019; Kazour et al., 2019; Mistri et al., 2018; Sonke et al., 2024; Zhu et al., 2024). The Mediterranean sea portion of the georeferenced global model estimations for shelf P by Cau et al. (2024) and Zhu et al. (2024) were isolated using QGIS. The rest of the literature concentrations for plastic in sediments are expressed in terms of the number of plastic items per g of dry sediment (n [item g$^{-1}$ DW]). These reported concentrations were converted into plastic mass per volume of





wet sediment ($q$ [g m$^{-3}$ WW]) using Eq. (5):

$$q = n \frac{L^3}{10} \, \delta_P \, \delta_S \, \tau_S \tag{5}$$

$L$ [cm] is the characteristic length of the plastic particles, $\delta_P = 1$ g cm$^{-3}$ and $\delta_S = 1.70$ g cm$^{-3}$ are the average density of plastic particles (Kooi and Koelmans, 2019) and marine sediment (Tenzer and Gladkikh, 2014) respectively, and $\tau_S = 2.97$ [-] is the wet to dry sediment density ratio (Barrett et al., 2020). This volumic concentration $q$ is then converted into a sea floor surface concentration [g m$^{-2}$] by estimating the mass of sediments deposited since 1950, based on sedimentation rates (see Eq. (S1) for details). This method accounts for the variation of plastic sedimentation rate since 1950, and for different core sampling depth. The corrected concentrations are then multiplied by the sea floor surface (4.50 10$^5$ km$^2$ for shelf sediment, 2.05 10$^6$ km$^2$ for deep sediment (Poulos, 2020)) to obtain the total stock for the compartment. A single study by Cutroneo et al. (2022) observed LMP concentration in the deep Mediterranean sediments. However, this study is judged not representative of the whole mediterranean deep sediment pool as it was conducted in a marine canyon located 10 km from the city of Toulon (France), and therefore very impacted by coastal human activities. A summary of the literature data on shelf sediments is presented in Table 2. Estimated stocks are $1.0 \pm 0.7$ Mt and $21.3 \pm 36.8$ Mt for shelf sedimented P and LMP respectively.

### 2.4.5 SMP extrapolation from global data

Since no reliable data on SMP stocks were available at the time of this study for the Mediterranean Sea, we use a first approximation based on a similar global modelling study by Sonke et al. (2024). The extrapolation was done by surface (Sea surface, Sandy beach, Shelf sediment) or by volume (Water column) and results are presented in Table 3.

**Table 3.** (single column) SMP extrapolation from Sonke et al. (2024) global study. Values are taken for the year 2015. Uncertainties are mean $\pm 1\sigma$.

| Compartment | Global estimate [Mt] | Mediterranean extrapolation [Mt] |
|---|---|---|
| Sea surface | $0.11 \pm 0.05$ | 7.6 10$^{-4}$ $\pm$ 3.5 10$^{-4}$ |
| Sandy beach | $0.20 \pm 0.19$ | 8.3 10$^{-4}$ $\pm$ 7.9 10$^{-4}$ |
| Water column | $41 \pm 17$ | $0.12 \pm 0.05$ |
| Shelf sediment | $14 \pm 6.7$ | $0.22 \pm 0.10$ |

### 2.4.6 Runoff to sea surface

Rivers play an essential role in the large-scale transport of plastics from terrestrial compartments to the ocean (Meijer et al., 2021). Despite being studied extensively, through field and modelling work, the uncertainties around the magnitude of the river leakage pathway remain several orders of magnitude (González-Fernández et al., 2023). We reviewed 8 modelling studies on plastic runoff from land to sea, 6 of them calibrated with observed plastic fluxes (Boucher and Billard, 2020; Kaandorp et al., 2020; Lebreton et al., 2017; Mai et al., 2023; Meijer et al., 2021; Schmidt et al., 2017), and the remaining 2 using human



activity and hydrology data (coastal population, MMPW data, river flow) (Jambeck et al., 2015; Nyberg et al., 2023). Among the 8 studies reviewed, only 2 focused exclusively on the Mediterranean region (Boucher and Billard, 2020; Kaandorp et al., 2020), while the other studies were calibrated globally and their results isolated for the mediterranean region. Results of this review are presented in Table 4.

**Table 4.** (single column) Literature review of plastic runoff to the Mediterranean Sea surface. Ptot = P+LMP+SMP, P $\geq$ 5mm, 0.3mm $\leq$ LMP < 5mm and $1\mu$m $\leq$ SMP < 0.3mm.

| Reference | Plastic runoff to sea surface [Mt y$^{-1}$] | | |
|---|---|---|---|
| | Ptot | P | LMP |
| Boucher and Billard (2020) | 0.23 (0.15 - 0.61) | 0.22 (0.14 - 0.58) | 0.013 (0.004 - 0.033) |
| Jambeck et al. (2015) | 0.65 (0.39-1.04) | | |
| Lebreton et al. (2017) | 0.0013 (0.0007-0.0032) | | |
| Mai et al. (2023) | | 0.0019 | |
| Meijer et al. (2021) | | 0.030 | |
| Nyberg et al. (2023) | | 1.24 | |
| Kaandorp et al. (2020) | 0.0026 (0.0021 - 0.0034) | | |
| Schmidt et al. (2017) | 0.054 (0.015 - 0.093) | 0.0075 | 0.047 (0.007 - 0.086) |
| Median (IQR 25-75%) | 0.05 (0.0026 - 0.23) | 0.03 (0.0075 - 0.22) | 0.03 |

To estimate the relative contribution of each region to the total plastic runoff from and to sea, the observations from Cózar et al. (2024) were used. This study links floating marine litter (mostly macroplastic) aggregates at the Mediterranean Sea surface, visible through satellite imaging, to land-based sources. The geographical distribution of plastic runoff was found to be dominated by S. Europe (87.9%), followed by N. Africa & M. East (12.0%) and the Nile basin (0.1%). These results are in contradiction with several earlier studies that estimated the geographical distribution of plastic runoff globally (Lebreton et al., 2017; Meijer et al., 2021; Nyberg et al., 2023). For each of these 3 studies, we extracted the Mediterranean data from the global dataset, and the relative contribution of each region computed in QGIS (version 3.28.4). Table 5 is summarizing the different results. The studies from Lebreton et al. (2017) and Meijer et al. (2021) are calibrated on a global dataset, which include limited observations from Mediterranean rivers, and do not include any African rivers. The model from Nyberg et al. (2023) is not calibrated against observed data. For these reasons, we consider that these 3 earlier studies are less robust in representing the geographical repartition of plastic runoff at the Mediterranean scale. We therefore decided to adopt the Cózar et al. (2024) results on macroplastic distribution, as it is specifically based on Mediterranean Sea observations. A detailed decision table supporting this choice is presented in Table S3. In the absence of other reliable estimations on LMP and SMP runoff distribution, the Cózar et al. (2024) percentages (Table 5) were applied to LMP and SMP for computing plastic runoff to sea in Eq. (6).



**Table 5.** (single column) Estimation of the relative contribution of Mediterranean regions to the total plastic runoff to the sea.

| Sources | Total plastic runoff [Mt y⁻¹] | S. Europe | N. Africa & M. East | Nile basin |
|---|---|---|---|---|
| Lebreton et al. (2017) | 0.001 | 16.3% | 60.1% | 23.7% |
| Meijer et al. (2021) | 0.023 | 1.0% | 78.5% | 10.5% |
| Nyberg et al. (2023) | 1.240 | 5.6% | 82.3% | 12.1% |
| **Cózar et al. (2024)** | - | **87.9%** | **12.0%** | **0.1%** |

## 2.5 Model calibration

To calibrate the model, the mass transfer coefficients (set of all k-values) have to be determined. In this study, most of the k-values were fitted to the median of the observations (See Fig.4). All the SMP k-values (except for continental runoff) were imported from the Sonke et al. (2024) global box-model. We acknowledge that the behaviour of SMP might be significantly
different between global and regional scale, and therefore consider this study as a first approximation. We recall that SMP is an important size category to incorporate into our outlook, despite the uncertainties, since their potential toxicity is a major concern for the environment. The coefficients for terrestrial runoff from the MMPW pool to the Mediterranean Sea surface are computed using Eq. (6).

$$k_{runoff}(R,P) = f_{runoff}(R) \frac{F_{runoff}(P, t = 2015)}{M_{mism.}(R, t = 2015)} \tag{6}$$

$M_{MMPW}$ [Mt] is the estimated terrestrial MMPW stock in the region R in 2015 (estimated after a single model iteration), and $f_{runoff}$ [-] is the relative contribution of each region to the total plastic runoff (see Table 5). The total plastic runoff for each size category Frunoff [Mt y⁻¹] is fitted to match the corresponding plastic stock in the marine environment. Uncertainties for plastic runoff were simulated based on the IQR of the most uncertain box (LMP in shelf sediment). To avoid unreasonably large uncertainties on well documented compartments such as P and LMP at the sea surface, the sinking k-value from sea
surface to shelf sediment is set to be proportional to the total LMP runoff from the MMPW terrestrial pool. As such, extreme runoff of LMP to the sea surface would be compensated by a corresponding change in LMP sinking.

## 2.6 Model scenarios

Following model calibration, we explore different plastic production and waste management scenarios to estimate the evolution of environmental plastic stocks and fluxes. Three OECD and one remediation scenario were simulated from 1950 to 2060 (and
240 beyond, to 2100 keeping OECD statistics fixed after 2060), from low to high ambition (OECD, 2022):

❖ **Business-As-Usual (BAU):** The BAU scenario is a simple projection of the historical plastic waste generation toward 2060. This scenario assumes that no further policies restricting plastic usage will be adopted, and accounts for the increase in population and income. BAU predicts that plastic production and waste generation roughly triples by 2060. Note that this scenario accounts for improvements in recycling fraction (from 11% in 2015 to 27% in 2060 for S. Europe,





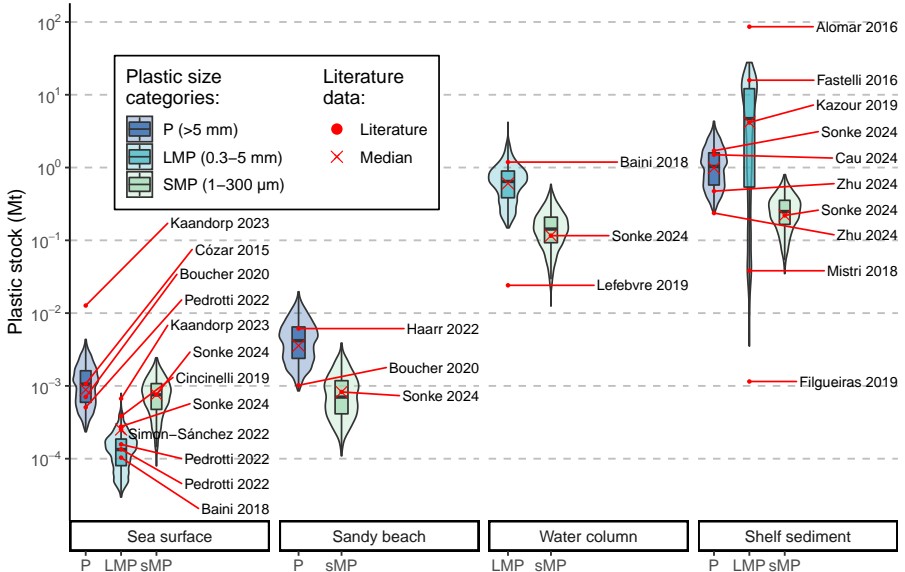

**Figure 4.** (double column) Model calibration for the year 2015. The Monte-Carlo model output distribution ($N = 1000$ iterations) for the year 2015 are represented with violin and boxplots, and compared with the literature data (red dots). The median of the literature data points is represented by a red cross. Only boxes with corresponding literature data are plotted. Note that the y-axis is log-scaled and common to all the compartments.

and from 5% in 2015 to 10% in 2060 for N. Africa & M. East and the Nile basin) and lowering of mismanaged fractions in all regions (from 7% in 2015 to >1% in 2060 for S. Europe, and from 53% in 2015 to 34% in 2060 for N. Africa & M. East and the Nile basin, see Fig.5).

❖ **Regional Action (OECD-RA):** This scenario consists of a global reduction of 18 % of plastic waste generation compared to the 2060 BAU projection, linearly increasing from 2025. The RAS also sets recycling goals toward 2060 for different economic regions: 40% recycling rate for EU countries (assimilated to S. Europe) and 20% for other countries (assimilated to N. Africa & M. East and the Nile basin) by 2060, linearly increasing from 2025. Similarly, OECD-RA aims to reduce the fraction of MMPW by 2060 to <1% in S. Europe and 7% for N. Africa & M. East and the Nile basin (see Fig.5).

❖ **Global Ambition (OECD-GA):** More ambitious than the OECD-RA scenario, it aims to enhance recycling rate to 80% by 2060 for EU countries (assimilated to S. Europe), and 60 % for N. Africa & M. East and the Nile basin. Additionally, OECD-RA aims to reduce the fraction of MMPW by 2060 to <1% for all regions (see Fig.5). This scenario also aims to reduce plastic waste generation by 33% globally compared to the 2060 BAU projection.

❖ **Remediation:** Alongside OECD-GA measures, we propose a remediation scenario where macroplastic, P, from the MMPW terrestrial pool is removed and stored in sanitary landfills where we suppose no leakage to the rest of the envi-





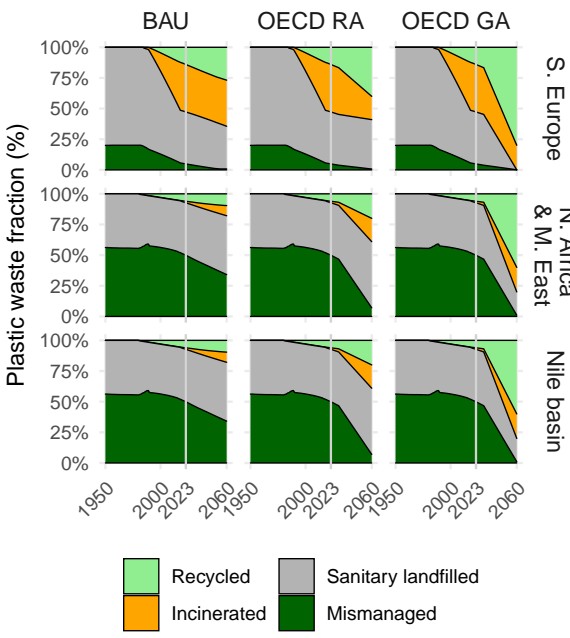

**Figure 5.** (single column) Fate of plastic waste for each scenario modelled. Data from OECD report (OECD, 2022). BAU: Business As Usual scenario, OECD-RA: Regional Action scenario, OECD-GA: Global Ambition scenario. The Remediation scenario uses the same plastic waste fate parameters as OECD-GA, it is therefore not represented here.

ronment. This can be achieved by dedicated waste management and remediation efforts, dedicated cleaning campaigns, and by increasing education and awareness of populations. Several options for this remediation are possible, but we arbitrarily choose to set an objective of 25% remediation of the MMPW P pool per year by 2060, linearly increasing from 2023 for all 3 regions.

# 3    Results and Discussion

We first present and discuss the modern Mediterranean plastic budget for 2015, with key metrics based on production and waste statistics, environmental plastics observations and model estimates (Fig.2). We recall that model fluxes are optimized to fit median plastic stock observations (Fig.4).

Based on OECD statistics, $476 \pm 24$ million metric tons (Mt) of plastic waste were generated in the Mediterranean Sea basin between 1950 and 2015, of which $21 \pm 1$ Mt was recycled (4%), $55 \pm 4$ Mt was incinerated (12%), $290 \pm 16$ Mt was

270 buried in sanitary landfills (61%), $110 \pm 17$ Mt was mismanaged (23%). Based on OECD statistics, we estimate that the region which generates the most plastic waste yearly is S. Europe with 67% (14 Mt y$^{-1}$) of the total plastic waste generation (21 Mt y$^{-1}$), followed by N. Africa and M. East with 19% (4 Mt y$^{-1}$) and the Nile Basin with 14% (3 Mt y$^{-1}$). We estimate that 7





Mt (median, IQR 3-15 Mt) of plastic is estimated to be present in the marine environment (Sea surface, water column, sandy beaches, shelf and deep sediments) which represents 1.5% of the total plastic waste generated in the Mediterranean Sea basin since 1950. This corresponds to a total plastic runoff from land to sea of 0.31 Mt y$^{-1}$ (median, IQR 0.14-0.57 Mt y$^{-1}$) in 2015. Following Cózar et al. (2024), the region contributing the most to plastic runoff to sea is S. Europe with 88% (0.27 Mt y$^{-1}$) of the total runoff, followed by N. Africa & M. East with 12% (0.04 Mt y$^{-1}$) and the Nile Basin with 0.1% (0.0003 Mt y$^{-1}$). Based on our observation calibrated model, we estimate that 83% of the plastic mass in the Mediterranean marine environment is concentrated in the shelf sediments (median 6 Mt, IQR 2-14 Mt). The Mediterranean Sea surface only contains 0.03% (median 2.0 10$^{-3}$ Mt, IQR 1.5 10$^{-3}$-2.7 10$^{-3}$) of the total mass of marine plastic.

Results on the atmospheric SMP cycle are not presented here, as they are not yet constrained enough in the Mediterranean region to provide reasonable levels of uncertainty.

## 3.1 Demographic trends toward 2060

Before discussing modelled plastic dispersal from present-day to 2060, we first summarize important demographic trends in the Mediterranean basin that will influence MMPW generation and runoff to the sea. The population in the Mediterranean catchment is expected to increase by 44% by 2060 compared to present values (540 M people in 2023 to 775 M people in 2060 ; UN (2022); WorldPop (2020)). This increase is mainly attributed to countries in the Nile basin (see Fig.3 and Fig. S3), with an additional 200 million people in less than 40 years. Combining population statistics (UN, 2022; WorldPop, 2020) and OECD MMPW per capita data (OECD, 2022), we calculare that the average Mediterranean catchment population generates around 10 kg of MMPW per year and per person (7 kg y$^{-1}$ capita$^{-1}$ for S. Europe and the Nile basin, and 20 kg y$^{-1}$ capita$^{-1}$ for N. Africa & M. East ; OECD (2022)), which is similar to our estimated global average (9 kg y$^{-1}$ capita$^{-1}$). The average per capita plastic runoff to the Mediterranean Sea is 0.7 kg y$^{-1}$ capita$^{-1}$ (S. Europe: 2.4 kg y$^{-1}$ capita$^{-1}$, N. Africa & M. East: 0.4 kg y$^{-1}$ capita$^{-1}$; Nile basin: 0.002 kg y$^{-1}$ capita$^{-1}$, Global: 2.2 kg y$^{-1}$ capita$^{-1}$). S. Europe population has a low MMPW generation per capita, but a high plastic runoff to sea per capita. On the contrary, N. Africa & M. East has high MMPW generation per capita and low plastic runoff to sea per capita. The S. Europe and N. Africa & M. East have similar population density and similar distance from the coast, which suggests that the difference in per capita contribution to the plastic runoff to sea is mainly due to other factors, in particular higher rainfall and associated flood events in S. Europe (Cózar et al., 2024). Despite hosting the largest population in the whole Mediterranean catchment (51% in 2023) and having high MMPW generation rates (about half of all plastic waste generated are mismanaged in 2023), the Nile basin has low contributions to both MMPW generation per capita (0.7 kg y$^{-1}$ capita$^{-1}$, similar to S. Europe) and per capita plastic runoff to sea (0.002 kg y$^{-1}$ capita$^{-1}$, 1 000 time less than S. Europe). This is probably due to the great length of the Nile river ( 6 700 km) and the numerous dams, known to trap MP (Watkins et al., 2019) and likely macroplastics, that have been constructed along its course. All per capita data, for the years 2023 and 2060 under BAU are available in Table S4.





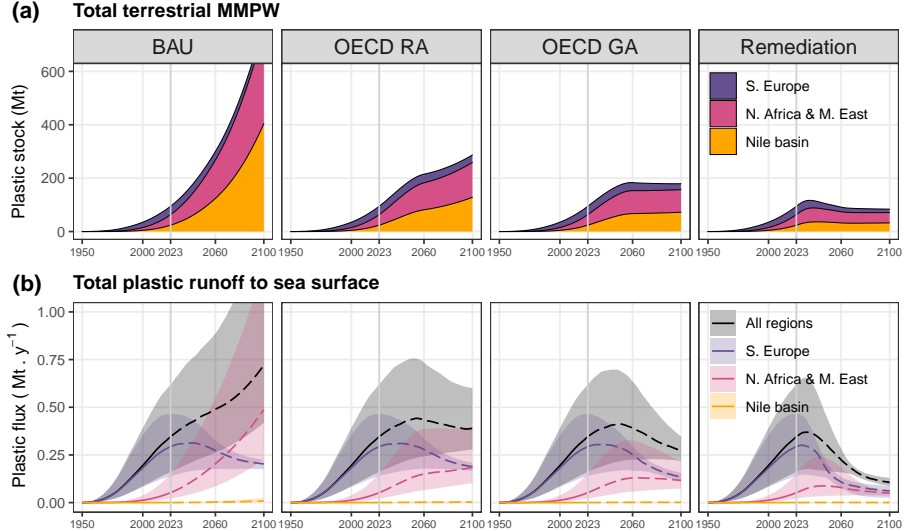

**Figure 6.** (double column) Total terrestrial mismanaged plastic waste (a) and total plastic runoff to the Mediterranean Sea surface (b) according to the four modelled scenarios: Business As Usual scenario (BAU), Regional Action scenario from OECD (OECD-RA), Global Ambition scenario from OECD (OECD-GA) and OECD-GA + terrestrial removal of mismanaged P (Remediation). See Fig.1 for region delimitation.

### 3.2 Total plastic waste generation and terrestrial MMPW

Figure 6.a and 7.a shows the amount and fate of total plastic waste generated in the Mediterranean basin based on OECD statistics (OECD, 2022). Under the BAU scenario, plastic waste generation is expected to increase by 190% by 2060, and by more than 400% by 2100, due to the increase in population and plastic consumption in the Nile basin. In OECD policy scenarios, emphasis is put on enhancing the recycled fraction (+320% in OECD-RA and +530% in OECD-GA, averaged in the whole Mediterranean region) and reducing the mismanaged fraction (-80% in OECD-RA and -97% in OECD-GA) by 2060.

Total plastic waste generation is also expected to decrease by 2060 under OECD scenarios, from 60 Mt y$^{-1}$ (2060, BAU) to 50 Mt y$^{-1}$ (2060, OECD-RA) to 40 Mt y$^{-1}$ (2060, OECD-GA). This however means that plastic waste generation will at least increase by 90% by 2060 compared to 2015 (21 Mt y$^{-1}$) under OECD scenarios.

Under the BAU scenario, the modelled terrestrial MMPW stock increases by more than 330%, from 72 Mt in 2015 to 310 Mt in 2060. This rise is mainly due to an increase in MMPW quantity in N. Africa & M. East (+430%, Fig.6.b) and the Nile

basin (+730%). The model also shows that efforts into reducing the mismanaged fraction implemented in OECD scenarios would allow this stock to be lowered to 220 Mt under OECD-RA, or to stabilize around 180 Mt under OECD-GA by 2060. We estimate that the implementation of remediation measures to remove P from this pool will further help reduce it to 90 Mt (P: 3%; LMP: 34%; SMP: 62%) by 2060. The modelled terrestrial remediation scenario would require a considerable cleaning effort: 2.7 Mt of mismanaged terrestrial P removed per year on average, for the period 2025-2060. This effort will peak to




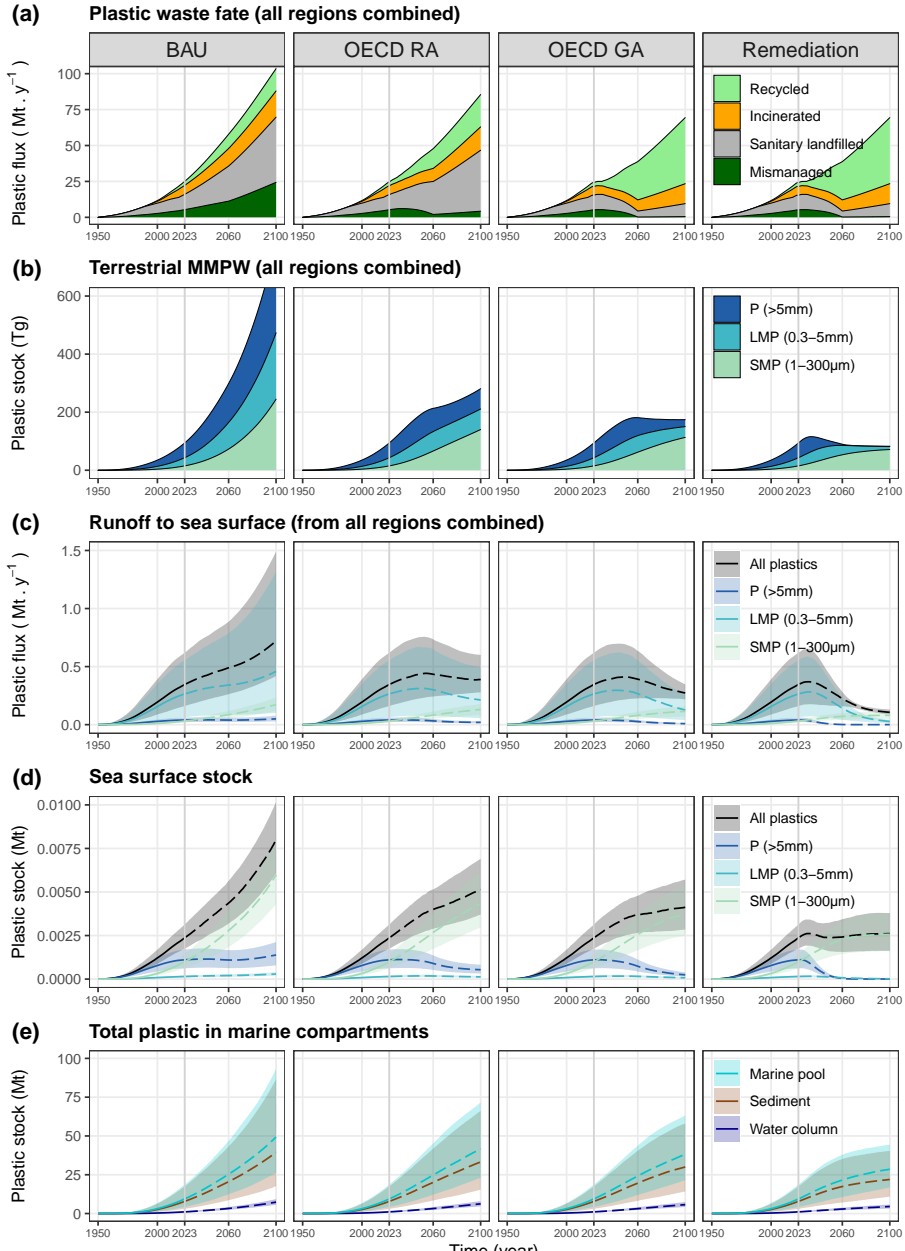

**Figure 7.** (double column) Comparison of the plastic transport to the marine environment between the four modelled scenarios: total plastic waste generation and fate (a), total terrestrial mismanaged plastic waste (b), plastic runoff to sea surface (c), Mediterranean Sea surface stock (d) and marine total plastic stock (e), including water column and total sediment (shelf + deep sediments). P: Macroplastics (>5mm), LMP: Large Microplastics (0.3-5mm) and SMP: Small Microplastics (1-300μm). BAU: Business As Usual scenario, OECD-RA: Regional Action scenario from OECD, OECD-GA: Global Ambition scenario, Remediation: removal of P in the MMPW terrestrial pool, alongside OECD-GA measures, with an objective of 25% remediation per year in 2060, increasing linearly from 2025. Uncertainties are Interquartile range (Q25-Q75).





a removal of 4 Mt y$^{-1}$ around 2040, which is very substantial: roughly 400 times superior to current (2023) global cleaning efforts by The Ocean Cleanup on rivers. However, this effort would be, at its maximum, equivalent to a daily removal of 17 g of mismanaged macroplastic per person, which is about half the weight of a standard 1.5 L plastic bottle. On average, this value would be 11 g capita$^{-1}$ d$^{-1}$ (roughly the weight of 2 credit cards).

### 3.3   Plastic runoff to sea

The simulated total plastic runoff to the Mediterranean sea is presented in Fig.7.c, and under the BAU scenario, is expected to increase by 40% by 2060 compared to present values, reaching 0.49 Mt y$^{-1}$ (0.35 Mt y$^{-1}$ in 2023). The OECD-RA scenario stabilizes total plastic runoff to 0.43 Mt y$^{-1}$ and OECD-GA allows a decrease of plastic runoff below the 2023 value by the end of the century. The Remediation scenario would lower plastic runoff to sea below the 2023 value by 2050, to stabilize around 0.1 Mt y$^{-1}$ by the end of the century.

In the model the total river plastic runoff of 0.31 Mt y$^{-1}$ (median, IQR 0.14-0.57 Mt y$^{-1}$) is constrained by the amount of total plastic observed in the marine environment. Our estimate is 3 times higher than the upper limit proposed by the Mediterranean Lagrangian model of Kaandorp et al. (2020) (0.1 Mt y$^{-1}$ for 2015, Fig.8). These authors did not use sediment plastic mass as a constraint on their plastic runoff estimate. However, this reservoir has considerable uncertainty (5 orders of magnitudes), which directly influences the runoff estimation. A more reliable estimate of plastic mass stored in sediments would help to 335  better constrain the model river runoff plastic flux. Our total plastic runoff to the Mediterranean sea is in accordance with the estimation of Boucher and Billard (2020) (0.23 Mt y$^{-1}$) and Jambeck et al. (2015) (0.65 Mt y$^{-1}$), which are among the highest estimations in the literature.

    The regional distribution of plastic runoff was constrained in our model using Cózar et al. (2024) georeferenced observation of near-shore floating debris. Cózar et al.'s observations suggest a different geographical plastic runoff distribution (S. Europe 340  » N. Africa & M. East » Nile basin) compared to previous studies that estimated Mediterranean plastic runoff based on global river observation (N. Africa & M. East > S. Europe   Nile basin) (Table 5). I this study, we privileged the observations from Cózar et al. (2024) over the older extrapolations, as discussed in the Method section (see Section 2.4.6 and Table S3). However, we emphasize that model trajectories of future marine plastic dispersal are very sensitive to the assigned geographical distribution of plastic runoff to the sea. This highlights the need for further research to reliably quantify regional contributions 345  to plastic runoff.

### 3.4   Plastic in the marine environment

    Modelled plastic stocks in the Mediterranean marine environment (sea surface, water column, sandy beaches, shelf and deep sediments) is presented in Fig.7.d-e. The total marine plastic pool is currently estimated to be 10 Mt (median, IQR 4-19 Mt), and could increase to 26 Mt (median, IQR 13-48 Mt) in 2060 under the BAU scenario. Both OECD-RA and OECD-GA would 350  have no significant effect on the total marine plastic stock by 2060 (Kruskal test, see Fig. S4). The Remediation scenario is the only scenario modelled here that could significantly lower the total marine plastic stock by 2060 to 22 Mt (median, IQR 11-38 Mt) which is still more than twice the current marine stock.



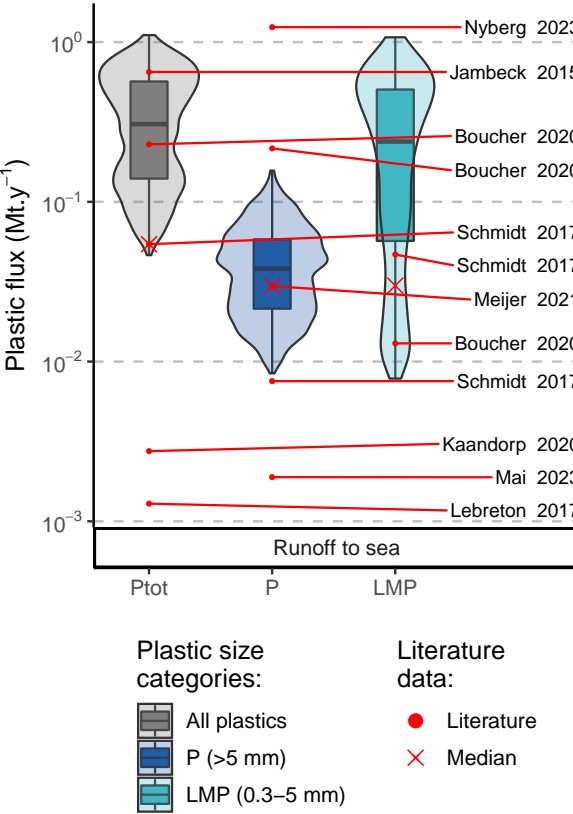

**Figure 8.** (single column) Comparison between modelled and literature plastic runoff to sea. The Monte-Carlo model output distribution ($N = 1000$ iterations) for the year 2015 are represented with violin and boxplots, and compared with the literature data (red dots). The median of the literature data points is represented by a red cross. Only boxes with corresponding literature data are plotted. Note that the y-axis is log-scaled.

The simulated Mediterranean Sea surface (Fig.7.d) would see its plastic stock increase by 83% between 2023 and 2060 under the BAU scenario (median 0.004 Mt, IQR 0.003-0.006 Mt), and potentially by more than 230% by the end of the century

(median 0.008 Mt, IQR 0.006-0.010 Mt). OECD scenarios seem to be efficient against P and LMP mitigation at the sea surface, but would still allow floating SMP stock to rise by 160% between 2023 and 2060 due to continuous fragmentation of legacy P and LMP on land, and subsequent SMP runoff to sea. The total floating plastic stock would reach 0.004 Mt under the OECD-GA scenario by 2060. Remediation of the MMPW terrestrial pool would allow to maintain total floating plastic stock close to the current estimate (median 0.0024 Mt, IQR 0.0018-0.0032 Mt).

It is important however to keep in mind that the sea surface compartment is the smallest stock of plastic in the marine environment. Most of the plastic runoff is expected to sink in the water column and deposit to sediments (Fig.7.e). By 2060, shelf sediments are expected to contain around 83% of total marine plastic. This forecast is concerning, as shelf sediments are





known to support most of the biodiversity of the Mediterranean Sea (Coll et al., 2010). Remediation of plastic stored in shelf sediments cannot be recommended, as these ecosystems are very fragile. This highlights the need to address plastic pollution
directly at the source, on land, and not only rely on remediation once plastic has spread to the marine environment. Likewise, the beached plastic stock is expected to double by 2060 and could potentially impact tourism, which is a major economical factor in the Mediterranean region.

## 3.5  Model limitations

Our modelling approach allows the exploration of various medium/long term policy scenarios, but is still subject to substantial
uncertainties. This work is meant as an exploratory approach, which could produce a general understanding for the dynamic of plastics in the environment, and highlight where data is lacking. Currently, the SMP cycle in the Mediterranean region is not constrained enough to allow any precise assessment. More observations of these smaller size fractions <300um are needed in all reservoirs, as SMP are a major threat to ecosystem and human health. Similarly, potential major pools of plastic (sediments, water column) are still very uncertain.

At this stage, we choose to ignore P in the water column and deep sediment compartments due to lack of observation. Nevertheless, we acknowledge that P can be transported by turbidity currents from shelf to deep sediment, and that macro litter from coastal sources and fishing activities can directly sink to the deep sediments.

Numerous improvements to this modelling work are necessary, and briefly summarized below. Fragmentation and biodegradation should normally be a function of polymer type, size and shape (Chamas et al., 2020). However, we do not consider
plastic polymer properties in the model, and the goal is to include this in future work. Important uncertainties in the mass budget come from the conversion of plastic number concentration to mass concentration. Indeed, the reported uncertainty on the mean size of plastic particles (L) is raised to the power 3 during the numeric to mass conversion ($V = L3/10$). To allow a more precise mass budget in the future, we recommend reporting plastic observations in both number and mass concentration. Future model development should aim to provide number concentrations, as this information is critical for ecotoxity risk
assessments.

Plastic stock and fluxes inside the study area are assessed in this study. Net plastic exchange between the study area and the rest of the globe (through human activities, water currents, by air or by land) is considered negligible. We acknowledge however that plastic waste may travel long distances by land before being recycled, incinerated, landfilled or mismanaged, and potentially leave the catchment area, or at the contrary, be imported inside the study region. We choose to ignore plastic
inflow from the North Atlantic via the Gibraltar Strait following Liubartseva et al. (2018) and Kaandorp et al. (2020). This study estimated leakage from discarded plastic pools to the environment, but these leaks represent only a few percent of all plastic waste generated since 1950 ( 3% according to Sonke et al. (2024),  2% in this study). This means that most of MMPW is still in terrestrial areas, which are not detailed yet in the model. Future improvements can be to separate urban, agricultural and natural soils or freshwater (lake, riverbed and floodplain) systems to better understand the fate of plastics on land. Finally,
the model does not yet account for plastic fragmentation to nanoplastics (NP). However, it is becoming apparent that NP also




causes serious risk for biota and human health (Atugoda et al., 2023). Literature addressing environmental NP concentrations are rapidly increasing, possibly allowing NP to be implemented in future versions of the model.

## 4  Conclusions

In this study, we explored the fate of plastic waste generated in the Mediterranean Sea catchment. We proposed a mass budget for the year 2015, and we calibrated a box-model to evaluate different OECD policy scenarios toward 2060. According to our estimates, the Mediterranean marine environment (sea surface, water column, sandy beach and sediments) stocked around 7 million metric tons (Mt) (median, IQR 3-15 Mt) of plastic in 2015. According to our model, this stock would increase almost 4-fold by 2060 under a business-as-usual scenario, reaching 26 Mt (median, IQR 13-48 Mt). Our results highlight the need to address plastic pollution directly on land, since >90% of the legacy MMPW is still terrestrial and will leak into the environment for decades and centuries to come. According to the OECD Global Ambition scenario modelled here, the plastic stock in the Mediterranean marine environments (sea surface, water column, beach and sediments) is expected to increase by 150% between 2023 and 2060, reaching 25 Mt. This estimation is not significantly different from the Business-as-Usual scenario, which suggest that more ambitious measures should be implemented, such as terrestrial remediation of MMPW. Current scenarios proposed by the OECD mainly impact the dispersion of macroplastics and large microplastics (>300$\mu$m). Small microplastics (1-300$\mu$m) will however be formed by fragmentation of larger plastic items, perpetuating the plastics cycle. Dedicated, large scale remediation of mismanaged terrestrial macroplastics could decrease, by the end of the century, the quantity of small microplastic in the marine environment by 36% compared to the Business-as-Usual scenario. Most of the marine plastics (83%) would accumulate in the shelf sediment, which concentrates most of the Mediterranean biodiversity.

In order to prevent escalating environmental impacts of plastics on the Mediterranean environment beyond present levels, ambitious policies are needed. Recommended measures are:

- ❖ Ambitious reduction of plastic waste generation (and by extension, virgin plastic production and consumption)
- ❖ Significantly reduce plastic leakage to the environment, by for instance promoting efficient collection systems and education.
- ❖ Encourage remediation of terrestrial legacy plastics. Upstream remediation and prevention are to be prioritized.
- ❖ International cooperation, as the Mediterranean Sea is surrounded by contrasting socioeconomic environments.

As international negotiations around a legally binding tool to address plastic pollution are unfolding (IUCN, 2023), an evaluation of the impact of different environmental policies on a regional and global scale is needed. Further modelling development would help to better apprehend the fate of plastic and microplastic in the environment, and identify efficient strategies to address this pressing issue. Our results highlight the need for more reliable observations in critical compartments such as marine sediments and subsurface waters to better constrain the plastic mass budget and reduce modelling uncertainties. Moreover, small microplastics (1-300$\mu$m) are not documented in the basin, which considerably increases the uncertainties regarding their



cycling, and fate. Additionally, more studies on plastic sinking, degradation and emissions in the environment would help to better constrain plastic fluxes and lower model uncertainties.

*Code and data availability.* Model code and supplementary data are available on: https://github.com/TheoSEGUR/Med_Plastic_BoxModel.
git

*Author contributions.* JS conceived the study. TS developed the model code and processed the data. TS conceived and wrote the manuscript, JS reviewed the maniscript.

*Competing interests.* The authors declare no competing financial or other interests.

*Acknowledgements.* We acknowledge financial support via the ANR-20-CE34-0014 ATMO-PLASTIC and ANR-23-CE34-0012 BUB-
BLPLAST grants, and a PhD scholarship from the French ministry of higher education and research.



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
