# Peer review of "TERRESTRIAL AND MARINE PLASTIC POLLUTION OUTLOOK IN THE MEDITERRANEAN REGION: A BOX-MODEL APPROACH BASED ON OECD POLICY SCENARIOS"

_EGUsphere, 2024_

## Author Comment (AC1)

This article presents a model of plastic pollution in the Mediterranean area. A mass budget for the year 2015 was proposed, and a box-model was calibrated to evaluate different OECD policy scenarios up to 2060. All steps and assumptions of the method are well-justified, and the results and conclusions are thoroughly developed. The authors critically examine the limitations of the current model and provide suggestions for improvement, although some of these potential improvements strongly depend on further studies for determining microplastics in the various environmental compartments considered. In particular, the lack of information on the concentration of nanoplastics in these compartments leads to a portion of the plastic pollution being 'missed,' an issue that must be addressed in future research. This type of study is crucial for understanding the potential impact of plastic waste reduction policies and for emphasizing the lack of ambition in current proposals. It is key to fostering the development of more robust and effective policies. This article is recommended for publication after minor revisions. The comments are provided below.

**Abstract**

Line 4: « based on recent observations. » be more specific, From what year is the literature used in this study?

The sampling period of all studies reviewed here range from 2013 to 2020 with an average sampling year of 2015. This year is therefore used as the reference for calibration. The publication date of all studies considered here range from 2015 to 2024. We added the comment "*The year 2015 is chosen as reference for calibration as it is the average sampling date of all studies reviewed here*" at L213.

**Introduction :**

**Line 30 :** Introduce the idea that plastic waste fragments in the environment, which in turn influences its mobility.

This is indeed a very important part of the dynamic of plastics in the environment, which was lacking in the introduction. We added this sentence to the introduction L35-38: "*Plastic objects in the environment fragment from macroplastics (>5mm) to microplastics (1µm – 5mm) to nanoplastics (<1µm) under the combined effects of (photo-)oxidation, structural fragilization and mechanical abrasion at a rate of approximately 3% per year (Chamas et al., 2020; Lebreton et al., 2019; Sonke et al., 2022, 2024).*"

Line 51 : Please specify what is referred to as plastic litter in this context. Later in the text, it appears to include macroplastics, but does it also encompass microplastics?

In this study, marine litter was used to be coherent with the sources quoted. Marine litter is defined by UNPE and IMO as human-created solid material released to the ocean, and includes indeed macroplastics and microplastics. We added this clarification to L69.

Lines 50-55 : What can explain the variability between studies, and how does your study address these uncertainties?

Generally, variability between observation studies can be related to several factors. Natural variability between sites can be very substantial and difficult to assess due to the lack of homogeneity in sampling and analytical methods. In this particular example, Pedrotti et al. (2022) used a Manta net (mesh size 333µm) and Cózar et al. (2015) used a Neuston net (mesh size 200µm), and their observed plastic size range is therefore similar. Both studies used a visual sorting method. These two studies differ by their extrapolation method to the whole Mediterranean Sea surface. Cózar et al. (2015) simply aggregated observation (after correcting for wind speed) from different datasets, while Pedrotti et al. (2022) used

a more elaborated Lagrangian model. This difference in extrapolation method, and the fact that the two studies used two different datasets, is likely explaining the difference between the two studies. Nevertheless, both studies provide estimates within the same order of magnitude.

In this study we address this uncertainty by reviewing more data. Across all the sea surface studies reviewed here, the uncertainty of sea surface plastic stock is also relatively low ("only" 2 orders of magnitudes between the lower and highest estimates according to Fig. 4), which is among the smallest uncertainties given the higher number of observations for the sea surface compartment.

During calibration, we address the uncertainties of the observed plastic stocks by transposing the k value uncertainty to the downstream box uncertainty. We added to clarify LXX. "*The uncertainties of the k-values were optimized to match the uncertainty of the literature observations in the corresponding downstream boxes*".

**Methods**

Lines 90-91 : « small microplastics (SMP) defined as 1 µm ≤ L<0.3 mm ». "Why this category? It is not a standardized size classification in the literature."

SMP (1 – 300 µm) were considered in this study following box model development in Sonke et al. (2022, 2025). The limit at 300 µm correspond both to the lower bound of plankton/neuston nets and to the approximate upper size of airborne microplastic able to travel between compartments (Shaw et al., 2023). We added this sentence at L132 to clarify "*The upper size limit of SMP correspond to the usual plankton net mesh size used to sample LMP in surface ocean. It is also a good estimate of the upper bound of airborne microplastics*"

Lines 96-97 : « Fragmentation is caused by (photo-)oxidation, structural fragilization and mechanical abrasion at a rate of approximately 3% per year (Chamas et al., 2020; Lebreton et al., 2019; Sonke et al., 2022, 2024) ». This information should already be presented in the introduction, see comment below

We moved this line to the introduction, L35. We then adjusted this sentence to avoid repetition with the introduction. L139 "Fragmentation rate was set to 3% per year following Chamas et al., 2020; Lebreton et al., 2019; and Sonke et al., 2024, 2022."

Tables S1 and S2 : How do you explain that some of your sd are 0.0?

Sorry, thanks for raising this issue. Most of the sd = 0 arrise from the fact that they were kept constant in the model because no reasonable estimations of the uncertainty were available for these parameters, such as regional runoff, remote soil fraction or shelf fraction (Table S1). We added a comment to the Table caption explaining this: "Whenever uncertainties were unavailable, they are set to zero, and parameters treated as constants." We also corrected the uncertainty of 3 k-values (k_LMP_surf_to_wcol, k_P_surf_to_sand and k_P_surf_to_ssed). The manuscript values and SI were updated without change in any conclusion, this update only affecting slightly some values.

Lines 191-195 : SMP extrapolation from global data. Could you please explain in more detail how you performed this extrapolation? Why is this the best option? Could an extrapolation from P and LMP in the Mediterranean be considered as an alternative?

Facing the lack on documentation about SMP in the Mediterranean Sea, we decided to use the same parameters (k-values) for SMP as Sonke et al. (2025). To keep consistency, we also choose to extrapolate SMP concentrations for the global ocean reviewed in Sonke et al. (2025).

We are currently working on a new improved method to extrapolate SMP from LMP that we will include in future modelling work. This approach uses plastic particle size distribution and should be more accurate than the method used here.

To clarify the extrapolation calculation, we added Table S3 and SI text VI the supporting information.

Lines 209-210 : It is more detailed, rather than being contradictory.

Indeed, the studies of Lebreton et al. (2017), Meijer et al. (2021) and Nyberg et al. (2023) are considering the plastic input to the ocean globally. Cózar et al. (2024) focussed specifically on the Mediterranean Sea. We changed "*contradictory*" by "*differ*" in L.314

**Results and Discussion**

-line 321 : It is important to note here that « The Ocean Cleanup » is a non-profit organization

We totally acknowledge that. We gave this comparison to put in perspective the work needed to achieve the goals set by the Remediation scenario, indicating that more than just non-profit organisation and NGO efforts are needed. We clarified in the text that The Ocean Cleanup is an NGO.

**References**

Cózar, A., Sanz-Martín, M., Martí, E., González-Gordillo, J. I., Ubeda, B., Gálvez, J. Á., Irigoien, X., and Duarte, C. M.: Plastic Accumulation in the Mediterranean Sea, PLoS ONE, 10, e0121762, https://doi.org/10.1371/journal.pone.0121762, 2015.

Cózar, A., Arias, M., Suaria, G., Viejo, J., Aliani, S., Koutroulis, A., Delaney, J., Bonnery, G., Macías, D., De Vries, R., Sumerot, R., Morales-Caselles, C., Turiel, A., González-Fernández, D., and Corradi, P.: Proof of concept for a new sensor to monitor marine litter from space, Nat Commun, 15, 4637, https://doi.org/10.1038/s41467-024-48674-7, 2024.

Lebreton, L. C. M., van der Zwet, J., Damsteeg, J.-W., Slat, B., Andrady, A., and Reisser, J.: River plastic emissions to the world's oceans, Nat Commun, 8, 15611, https://doi.org/10.1038/ncomms15611, 2017.

Meijer, L. J. J., van Emmerik, T., van der Ent, R., Schmidt, C., and Lebreton, L.: More than 1000 rivers account for 80% of global riverine plastic emissions into the ocean, Sci. Adv., 7, eaaz5803, https://doi.org/10.1126/sciadv.aaz5803, 2021.

Nyberg, B., Harris, P. T., Kane, I., and Maes, T.: Leaving a plastic legacy: Current and future scenarios for mismanaged plastic waste in rivers, Science of The Total Environment, 869, 161821, https://doi.org/10.1016/j.scitotenv.2023.161821, 2023.

Pedrotti, M. L., Lombard, F., Baudena, A., Galgani, F., Elineau, A., Petit, S., Henry, M., Troublé, R., Reverdin, G., Ser-Giacomi, E., Kedzierski, M., Boss, E., and Gorsky, G.: An integrative assessment of the plastic debris load in the Mediterranean Sea, Science of The Total Environment, 838, 155958, https://doi.org/10.1016/j.scitotenv.2022.155958, 2022.

Shaw, D. B., Li, Q., Nunes, J. K., and Deike, L.: Ocean emission of microplastic, PNAS Nexus, 2, pgad296, https://doi.org/10.1093/pnasnexus/pgad296, 2023.

Sonke, J. E., Koenig, A. M., Yakovenko, N., Hagelskjær, O., Margenat, H., Hansson, S. V., De Vleeschouwer, F., Magand, O., Le Roux, G., and Thomas, J. L.: A mass budget and box model of global plastics cycling, degradation and dispersal in the land-ocean-atmosphere system, Micropl.&Nanopl., 2, 28, https://doi.org/10.1186/s43591-022-00048-w, 2022.

Sonke, J. E., Koenig, A., Segur, T., and Yakovenko, N.: Global environmental plastic dispersal under OECD policy scenarios toward 2060, Science Advances, 11, eadu2396, https://doi.org/10.1126/sciadv.adu2396, 2025.

---

## Author Comment (AC2)

This study explored the fate of plastic waste generated in the Mediterranean Sea catchment and authors proposed a box-model to evaluate different OECD policy scenarios toward 2060. This is a highly forward-looking study, essential for the development of public policy. The model is well presented and results are interesting. I recommend this paper for publication after major revision.

Major comments and other comments

I recommend authors stating right from the introduction that this is a prospective, exploratory exercise, that it tries to take into account a maximum of plastic pollution (large, small, etc.), but that it cannot consider all categories for all the compartments but this model tries to take as much as he can. I appreciated the limitation of the model section, which really help to understand what the model proposes or not.

We added this sentence at the end of the introduction to clarify the nature and purpose of this study: L104. *"The aim of this exploratory study is to investigate the relevance of the OECD's scenarios regarding plastic pollution in the Mediterranean catchment, while taking into account a wide plastic size range (from microplastics to macroplastics) in a maximum of environmental compartments. This prospective assessment is a necessary step to guide public policies and decision makers, while UN negotiations for an international legally binding treaty to end plastic pollution are under way."*

My first and solely concern is the concept of runoff, which still unclear from my side. L70. of plastic runoff from land to sea. What do you mean by runoff form land-to-sea? Remobilization of all plastic litter undependably of the connection with river first and then Sea? Or direct runoff to the Sea? It's also parametrized to consider the intensity of the rain? A major conclusion of this study is the important of runoff – what I would call river discharges during high flow periods or during flooded but not definitely runoff. From my point of view, this concept, or what runoff mean, should be better explained.

To clarify this concept, we decided to replace the expression "runoff from land to sea" by "input from land to sea". The more generic term "input" is then defined on L298 as any leakage from the terrestrial plastic pool, including, to mention the more commonly quoted examples, rivers (law and high flow period), coastal urban and non-urban areas, fishing and aquaculture industries, and shipping activities. Literature on different leakage pathways were also quoted.

Implementation of the OECD Global Ambition policy scenario, that targets near-zero new plastics waste leakage, would not significantly lower this stock (25 Mt, median, IQR 12-44 Mt) by 2060. Totally agree with this conclusion. Very important as regard the OCED recommendation and the objectives. Behind this policy scenario, the main idea is to significantly reduce the plastic consumption and not necessary the plastic pollution in the environmental compartment. One other important idea is the "legacy stock" of plastic.

L15. his underlines the necessity to address upstream legacy plastic waste on land. How upstream? On land? Or for the consumption?

In this article, we refer to "*upstream legacy plastic waste*" as the terrestrial mismanaged plastic pool. Under the Remediation and OECD-GA scenarios, the mismanaged plastic production rates (fraction of the waste mismanaged in the environment) fall to near 0, which implies that the change in consumption, or not, won't affect new mismanaged plastic waste after 2060.

L24. Plastic items are also very mobile due to their relatively low density and buoyancy, and can travel long distances by rivers and ocean currents. Some studies suggest that significant amount of plastic litter can be also trapped along the river banks and flooded pain aera.

This is very true. The current model has no implementation of such compartment. The main reason is that its stock is not constrained enough as yet. Ultimately, the implementation of the terrestrial freshwater compartment in our model would allow a more temporally detailed approach, and for instance simulate seasonal river inputs to the sea. We would need new funding to develop this.

L69. This study investigates the fate of plastic waste in the Mediterranean Sea catchment across various environmental compartments 70 (terrestrial, sea surface and water column, shelf and deep sediments, beaches and atmosphere). Until here, we cannot determine if the study will focus on plastic litter only or include microplastic. If atmosphere is included, probably microplastic is considered. If yes, why? It's clearer L90-95.

To clarify this point, we replaced the generic term plastic in the introduction and in the abstract by '*macroplastics and microplastics (hereafter 'plastics')*' when referring to our model.

L280. Results on the atmospheric SMP cycle are not presented here, as they are not yet constrained enough in the Mediterranean. I suggest to remote this section to the core manuscript. It can be mentioned in model structure.

OK. We moved this comment to the section 2.4.5 SMP extrapolation from global data (L284)

Figure 2. Which data are used to build this figure?

Figure 2 display the modelled stock and fluxes at the beginning of year 2015. The scenario chosen is irrelevant, since they are all identical before 2015. We added the comment in the caption of Fig.2. "*for the year 2015 under all scenarios*" for clarification.

L112. Plastic mass transport between boxes is approximated to be first order, where the plastic flux $F_{A \to B}$ [Mt y-1] from box A to box B is proportional to the plastic mass $M_A$ [Mt] in box A. How can we justify this assumption? Which implication would have a different one? It's here a question of the assumption sensibility.

The assumption that fluxes between boxes (or between chemical compounds) follow first order rates (and equations) is justified by its wide application in chemical kinetics, radioactivity, and biogeochemical cycling. In all these systems, transformation or transport of matter is often linearly proportional to the amount of matter at the starting point. It is the simplest assumption one can make for upstream dependent fluxes in box models. First order fluxes have the advantage to need only one parameter (k-value) to be measured or optimized, which simplifies greatly the model calibration process against observations. The ideal observations to justify a first order assumption would be data on fluxes, for example from peat, sediment or ice core natural archives. These fields are in their infancy, though preliminary results show a gradual increase in historical MP deposition to such archives (Allen et al., 2021). A different assumption would be non-linear, higher order MP dispersion and fragmentation behaviour in the environment. Example could be that fragmentation rates accelerate as plastics age; or that climate change influences several fluxes (k values would become dependent on other factors), for example land to sea plastic inputs via changes in Mediterranean precipitation. Based on the absence of evidence of higher order behaviour we make the typical choice of the first order approach in our model.

We added a phrase in this sense to the cited section, on L164: "*Without broad evidence for higher order, non-linear plastic dispersion dynamics, we consider Eq.1 to be a reasonable assumption that is based on similar behaviour during chemical kinetics, radioactivity, and biogeochemical cycling.*"

Table 1. For each study, in addition to the stock considered, which compartments are considered? This is important since in your model you consider all compartments.

The compartment considered is Mediterranean Sea surface, as mentioned in the table title and column names.

2.4.6 Runoff to sea surface. Here what is the concept of runoff? It's the discharges by river during high flows/flooded?

As detailed above, we clarified the concept of what we considered runoff by changing our nomenclature and using input instead. Indeed, as you pointed at rightly, the term runoff has a specific meaning in hydrology that does not correspond exactly to our situation. In our case, inputs of plastic from land to sea include runoff, alongside a number of other processes such as direct littering in the sea, flood events, wind etc…

The geographical distribution of plastic runoff was found to be dominated by S. Europe (87.9%), followed by N. Africa & M. East (12.0%) and the Nile basin (0.1%). Do this observation is linked to the crossed explanation, population density and high flows of river? In contradiction with other studies (as underlined by authors). This has a very strong implications for policy. I would suggest to explain more why you found these contradictory results and how our approach is relevant in regards to others studies.

In this paper, we implement a bottom-up approach to estimate the total plastic input to sea: the observed quantity of plastic in the marine environment constrains the inputs from land. To be able to regionalize this flux between S. Europe, N; Africa & M. East and Nile basin, we use the % given by Cózar et al. (2024) are quoted in the sentence you mention. Cózar et al. derived this geographical distribution from satellite observations of marine litter. We only use population density to estimate the plastic waste generated in each region, and then calculate the mismanaged plastic waste using the % of MMPW provided by the OECD. We choose to select Cózar et al. (2024) results among all the other studies that investigate the geographical distribution of plastic input to sea because it was applied to the Mediterranean Sea specifically, and also adopted a bottom-up estimation. A recent study by Weiss et al., 2025 found similar fractions between our 3 regions (79.5% for S. Europe, 14.0% for N. Africa & M. East and 6.5% for the Nile basin). Their results are based on population density and river flow for each basin (top-down approach). We rewrote the paragraph at L.312 for clarity, and mentioned the study of Weiss et al. (2025) throw-out the text and figures.

L395. This means that most of MMPW is still in terrestrial areas, which are not detailed yet in the model. Totally agree.

Results section. Do the results/conclusions will be different by considering only the large fraction of plastic litter (> 5 mm)?

Most conclusions would be different because microplastic makes most of plastic inputs from land to sea, and because a non-negligible fraction of plastic waste are primary microplastics. Ignoring fragmentation of large plastic litter would also bias its mass budget, especially if microplastic that have deposited in sediment are ignored.

Minor comments

L4. for the Mediterranean region. Please clarify the aera.

We specified « Mediterranean catchment and Sea" for more clarity. Please also note that the next phrase specifies that "*Mediterranean watersheds in Southern Europe, Northern Africa and Middle-East, and Nile basin*" are considered.

L54. Simon-Sánchez et al. (2022) reviewed and reported concentrations in sediments (300 items kg-1) and beaches (60 item kg-1), insisting on the high uncertainties and 55 variability between studies. You discussed here about microplastics. What is the link with plastic litter mentioned?

This paragraph is a broad introduction to plastic pollution. The number quoted from Simon-Sánchez et al. (2022) are referring to all plastic items, macro and microplastics included.

L60. Sea litter by Cózar et al. (2024) highlighted the close relationship between marine litter occurrence and heavy rainfall events, pointing at Southern Europe as the largest macroplastic source to the Mediterranean Sea. What is their hypothesis?

Cózar et al. (2024) hypothesised that the cluster of marine litter observed at the sea surface by satellite imaging (referred to as *litter windrows*) are a good proxy for 'marine plastic litter'.

Based on the fact that plastic items represent as significant fraction of the total floating marine litter, Cózar et al. (2024) proposed to monitor litter windrows as a proxy for surface floating plastics. The reflectance spectra of plastic litter, alongside spectra of other confounding floating debris (algae, driftwood and seafoam) were compiled into a tool able to detect pixels containing litter windrows.

Importantly, this method is only a proof of concept, using already existing EU Copernicus Santinel-2 multispectral instrument that is suboptimal. The method is not yet able to detect litter windrows shorter then 70m, and is not able to estimate the percentage of plastic in these structures. The model only asses the presence or absence of dense litter cover in a 10*10m pixel.

Cózar et al. (2024) found a correlation between plastic input from land and litter windrow density, and particularly a correlation between rainstorm or flood events on land and the formation of litter windrows near the coast. They hypothesise that this phenomenon is the result of the flushing of the watersheds after high precipitation events. They indeed observe that southern Europe, despite presumed low mismanagement rate of plastics, is a major contributor to plastic input to the sea, as shown by the high concentration of windrows near its shore.

L142. The year 2015 is chosen as reference for calibration. Why?

The year 2015 was chosen as reference for calibration as it is the average sampling year of the reviewed literature. Also, this date was used in similar work on the same basis (Sonke et al., 2022, 2025), making direct comparison easier. We added this comment on L212. to clarify: "*The year 2015 is chosen as reference for calibration as it is the average sampling date of all studies reviewed here*"

Table 2. What is the unit of the first reported concentration? 5.6 10-3

We forgot to include the unit. The correct unit was "kg km$^{-2}$", fixed.

L289. we calculare. Calculate

Typo fixed

**References**

Allen, D., Allen, S., Le Roux, G., Simonneau, A., Galop, D., and Phoenix, V. R.: Temporal Archive of Atmospheric Microplastic Deposition Presented in Ombrotrophic Peat, Environ. Sci. Technol. Lett., 8, 954–960, https://doi.org/10.1021/acs.estlett.1c00697, 2021.

Cózar, A., Arias, M., Suaria, G., Viejo, J., Aliani, S., Koutroulis, A., Delaney, J., Bonnery, G., Macías, D., De Vries, R., Sumerot, R., Morales-Caselles, C., Turiel, A., González-Fernández, D., and Corradi, P.: Proof of concept for a new sensor to monitor marine litter from space, Nat Commun, 15, 4637, https://doi.org/10.1038/s41467-024-48674-7, 2024.

Simon-Sánchez, L., Grelaud, M., Franci, M., and Ziveri, P.: Are research methods shaping our understanding of microplastic pollution? A literature review on the seawater and sediment bodies of the Mediterranean Sea, Environmental Pollution, 292, 118275, https://doi.org/10.1016/j.envpol.2021.118275, 2022.

Sonke, J. E., Koenig, A. M., Yakovenko, N., Hagelskjær, O., Margenat, H., Hansson, S. V., De Vleeschouwer, F., Magand, O., Le Roux, G., and Thomas, J. L.: A mass budget and box model of global plastics cycling, degradation and dispersal in the land-ocean-atmosphere system, Micropl.&Nanopl., 2, 28, https://doi.org/10.1186/s43591-022-00048-w, 2022.

Sonke, J. E., Koenig, A., Segur, T., and Yakovenko, N.: Global environmental plastic dispersal under OECD policy scenarios toward 2060, Science Advances, 11, eadu2396, https://doi.org/10.1126/sciadv.adu2396, 2025.

Weiss, L., Estournel, C., Marsaleix, P., Mikolajczak, G., Constant, M., and Ludwig, W.: From source to sink: part 1—characterization and Lagrangian tracking of riverine microplastics in the Mediterranean Basin, Environ Sci Pollut Res, 32, 10081–10104, https://doi.org/10.1007/s11356-024-34635-6, 2025.